# Surface Nanocrystallization and Improvement of the Mechanical and Tribological Properties of AISI 304 Steel Using Multi-Pass Nanostructuring Burnishing

**DOI:** 10.3390/ma17225656

**Published:** 2024-11-20

**Authors:** Viktor Kuznetsov, Igor Tatarintsev, Vladimir Voropaev, Andrey Skorobogatov

**Affiliations:** 1Institute of New Materials and Technologies, Ural Federal University, 620062 Yekaterinburg, Russia; igor.tatarintsev@gmail.com (I.T.); sen_vvv@mail.ru (V.V.); ufo2log@gmail.com (A.S.); 2Engineering Center, Kurgan State University, 640002 Kurgan, Russia

**Keywords:** austenitic steel, nanostructuring burnishing, nanocrystalline structure, surface topography, microhardness, sliding speed, friction coefficient, wear resistance

## Abstract

Owing to their high producibility and resistance to corrosion, austenitic chromium–nickel steels are widely used in the chemical, petroleum, and food industries. However, their significant disadvantage lies in their poor structural performance, which cannot be improved by heat treatment. This significantly limits the usability of these steels in parts of machines that operate under friction loads. Hardening can be achieved by decreasing the size of grains and applying deformation-induced martensitic transformation. Nanostructuring burnishing (NSB) may be one of the technologies suited for producing parts of tribological assemblies with enhanced operating characteristics. Nanostructuring burnishing using a sliding indenter is being developed as a method of industrial surface nanocrystallization through severe plastic deformation used in the mechanical machining of various types of parts. This article investigates the possibility of enhancing the mechanical and tribological properties of nanocrystallized surfaces of austenitic steels, which are formed through nanostructuring burnishing using a tool with a natural diamond spherical indenter and a change in sliding speed from 40 to 280 m/min with one, three, and five passes. Increasing the tool sliding speed makes surface nanostructuring machining of big parts highly effective. This paper aims to establish the influence exerted by the sliding speed and number of indenter passes on the formation of a nanocrystalline structure, as well as on the modification of microhardness and residual stresses, texture, and tribological properties of the surface layer in the nanostructuring burnishing of AISI 304 steel. Transmission microscopy and microdurometry, 3D-profilometry, and tribological tests of surfaces nanocrystallized with the “ball-on-disk” scheme with dry and lubricated friction established the optimal values of speed and number of passes for a spherical indenter in nanostructuring burnishing.

## 1. Introduction

Owing to their high producibility and resistance to corrosion, austenitic chromium–nickel steels, such as AISI 304, are widely used in the chemical, oil and gas, and food industries. However, their significant disadvantage lies in their low structural performance, which cannot be improved by heat treatment. This significantly limits the usability of these steels in critical parts of machines that operate with elevated friction loads.

The hardness and working performance of such steels may be improved by applying functional coatings and chemithermal treatments. The mechanical properties and wear resistance of the surface layer may be improved to a high level by physical and mechanical machining, with the development of intensive plastic deformation resulting in the activation of dislocation and grain boundary strengthening mechanisms. The size of the grain structure is thus reduced to a nanocrystalline state, while the austenite undergoes deformation-induced martensitic transformation. The hardening and improvement of tribological properties of austenitic steels may be achieved directly during the mechanical machining of parts of tribological assemblies.

Study [1] investigates the improvement of the wear resistance of steel parts surfaces by applying modern functional coatings containing silicides of iron and other metals. Specifically, chromium silicide (CrSiN) has demonstrated prominent results. In study [2], a wear-resistant CrSiN-based coating was applied on a sample manufactured from AISI 304 steel. The thickness of the so-formed wear-resistant layer was 4.5 μm. A ball-on-disk tribological test established that the specific wear coefficient was 5 × 10^−16^ m^3^/(N·m), and the dry friction coefficient was 0.4. The authors of study [3] applied a tungsten silicide magnetron coating on a sample made of AISI 304 steel, which yielded a wear resistance coating with a thickness of 2 μm; a ball-on-disk tribological test established that the specific wear coefficient was 2 × 10^−15^ m^3^/(N·m).

Due to their high plasticity, robustness, and resistance to corrosion, coatings based on iron compounds are used to improve the wear resistance and tribological properties of parts. However, the disadvantages of traditional Fe-based coatings include low hardness and crack resistance, limiting their use under high loads or high wear intensity conditions [4,5]. In study [6], the authors analyzed a wear-resistant titanium carbide (TiC)–iron composite coating applied on an AISI 316L steel sample using the high-velocity air fuel (HVAF) process. The resulting coating had a thickness of up to 200 μm, with a surface roughness *Ra* of 6.2 μm and microhardness of 887 HV_0.03_. It was established that the dry friction coefficient was 0.41, and the specific wear coefficient was 5 × 10^−14^ m^3^/(N·m).

Structural characteristics of the AISI 304 steel surface can be improved using thermochemical treatment. M.A. Barchelos et al. [7] studied the technology of low-temperature plasma carburizing, which is based on the saturation of AISI 304 steel surface by carbon at temperatures below 500 °C, allowing the users to avoid the formation of carbides and preserve the austenitic structure of the steel, thereby improving its mechanical and tribological properties without decreasing the resistance to corrosion. This method hardened the surface layer of a workpiece made of AISI 304 steel at a depth of 3.9 to 11 µm. The maximum microhardness of the hardened surface layer was 475 HV_0.05_. The surface roughness after the machining was *Ra* = 0.2–0.3 µm, the dry friction coefficient in tribological tests was 0.6, and the specific wear coefficient was 1.2 × 10^−15^ m^3^/(N·m).

AISI 304 steel surface mechanical hardening methods were studied by D.A. de Oliveira, Xin Wang, F.C. Nascimento, M.A. Barcelos, and J. Dib et al.

Study [8] investigates the influence of the rolling velocity, number of passes, and contact pressure of the ball rolling tool in α′-martensite inducing surface deformation machining in AISI 304 steel. The highest content of α′-martensite was found near the surface, and it decreased in proportion with the layer depth. Employing a rolling velocity higher than 103 mm/rev decreases the content of α′-martensite, while the latter tends to increase as higher numbers of passes are used. The decrease in α′-martensite content, when the tool rolling velocity is increased, occurs due to the emission of adiabatic heat. This work also discovered that higher degrees of deformation with a higher number of tool passes result in an increase in the roughness and undulation of the surface. As pointed out by the authors, the higher wear resistance is achieved by forming martensite in the surface layer with a thickness of 0.255 to 2 µm, which increases the hardness of the surface layer in the range of 570–650 HV, depending on the number of passes and rolling velocity. This study does not specify the resulting roughness. However, study [9] demonstrates that after the low plasticity burnishing (LPB) of AISI 304 steel, the roughness was from 0.1 to 0.4 µm, depending on the process parameters. Higher feeding speeds, a lower number of passes, and lower contact pressure cause the roughness *Ra* to increase. Optimal roughness parameters are obtained when the feed rate is changed to 0.05–0.1 mm/rev, the normal force is changed to 100–200 N, and the number of tool indenter passes is changed to 2–3.

Working characteristics of AISI 304 steel can be significantly improved with surface machining by nitrogen–carbon glow discharge. Study [10] demonstrates that machining by nitrogen–carbon glow discharge was followed by the discovery of phases, such as residual austenite in the steel structure, which increases the structural performance of the surface; the formation of nitride phases, such as γ′-Fe_4_N and ε-Fe_2–3_N, results in the improvement of wear resistance. The microhardness of the surface layer at depths of up to 50 µm reached 3.5 GPa after machining, which is significantly higher than the initial values. This improvement is explained by the diffusion of nitrogen and carbon into the steel surface layer. The resulting surface roughness after the machining was *Ra* = 0.2 µm, and the dry friction coefficient of the sample machined surface was 0.5–0.7. Although this study does not provide the specific wear coefficient values, it states that the wear resistance improved by 5.8% after machining, compared with a non-machined surface.

Methods for the mechanical hardening of AISI 304 steel surface include nanocrystallization with intensive plastic deformation using compression and simple shear. The study by X.Y. Wang [11] proposes utilizing sand blasting combined with annealing at 350 °C for 3 h for the nanocrystallization of the surface layer using severe plastic deformation. A nanocrystalline structure with crystals up to 20 nm in size was formed on the AISI 304 steel surface layer. The surface layer depth was up to 70 µm. A microhardness of over 400 HV was found at a depth of the hardened layer of less than 30 µm. At a distance from the surface of more than 30 µm, the microhardness gradually decreased and reached initial pre-modification values at a depth of 70 µm. After the nanostructuring process, the sample surface was machined by polishing to obtain a roughness of *Ra* = 0.05 µm. Further, this paper provides the results of tribological tests, establishing that the dry friction coefficient was 0.72, and the lubricated friction coefficient was not more than 0.11. The specific wear coefficient was 1.13 × 10^−14^ m^3^/(N·m).

In study [12], the hardening of the AISI 304 steel surface layer was achieved by polishing the surface with silicon carbide, consequently using abrasive cloths 60, 120, 320, 400, and 600. This process increased the martensite volume on the surface. The microhardness of the resulting surface at a depth of up to 50 µm was up to 420 HV_0.05_. The surface roughness *Ra* was 0.035 µm, the dry friction coefficient of the surface was 0.4–1.2, the friction coefficient under lubricated friction was 0.15, and the specific wear coefficient was 3.38 × 10^−14^ m^3^/(N·m).

Study [13] investigated the impact of the parameters of AISI 304 steel surface texture on its tribological properties using dry friction. The study established a correlation among the arithmetical mean height of the scale-limited surface (*Sa*), the autocorrelation length (*Sal*), the reduced peak height (*Spk*), and the AISI 304 steel dry friction coefficient. It discovered that smoother and more even surfaces with low *Sa*, *Sal*, and *Spk* parameter values demonstrate lower dry friction coefficients.

Study [14] investigated the impact of the texture parameters of steel surfaces (defined by ISO 25178-2) on their tribological properties. This study discovered that the dry friction coefficient correlates to the highest extent of the following parameters: *Spk*, *Sa*, the developed interfacial area ratio of the scale-limited surface (*Sdr*), and the root mean square gradient of the scale-limited surface (*Sdq*).

The authors of study [15] established that surfaces with low dale void volume of the scale-limited surface (*Vvv*) parameter values are characterized by higher friction coefficients under lubricated conditions, which is explained by their low oil-retaining capacities.

Review article [16] addresses the wear of AISI 304, 321, and 316L steels after surface nanocrystallization with surface mechanical attrition treatment (SMAT), shot peening (SP), sandblasting, supersonic fine particle bombardment (SFPB), fast multiple rotation rolling (FMRR), high-speed pounding (HSP), ultrasonic impact treatment(UIT), ultrasonic surface rolling (USR) process, and surface mechanical rolling treatment (SMRT) and proposes wear resistance improvement mechanisms. The formation of surface roughness after nanostructuring machining and its impact on the quality and tribological properties of the surface were not sufficiently studied. The authors also recommend studying the impact of the grain size gradient on the tribological properties of the material.

In study [17], researchers observed an increase in the volume of forming deformation martensite on the surface of AISI 304 austenitic steel with an increase in the ball size and duration of the SMAT machining. Specifically, the number of shot blasting cycles significantly influenced the volume of deformation martensite forming on the surface of AISI 304 steel and the depth of the layer that undergoes deformation-induced martensitic transformation [18]. Shot blasting and SMAT machining increased the microhardness of AISI 304 steel by 2.5 times [19,20].

One of the methods of the nanocrystallization of steel surface layers is nanostructuring friction burnishing using a sliding indenter. Nanostructuring burnishing using a spherical or cylindrical indenter resulted in the simultaneous formation of a nanocrystalline structure and the flattening of the initial surface microprofile to obtain nanotopography. The technological impact on the surface layer depends on the normal force *F_b_*, sliding speed *v_s_*, and feed *f_b_* of the indenter, as well as the number of working passes *n_p_*, friction coefficient *µ*, and radius *R* of the tool’s working part. Each of the listed parameters has a significant influence on the parameters of the severe plastic deformation of the material, nanocrystallization, and the properties of the surface layer [21,22].

Study [23] by V.P. Kuznetsov et al. provides the results of experimental studies of AISI 304 steel after finish turning and nanostructuring burnishing with one and five passes of a diamond spherical indenter with a 2 mm radius. This research demonstrates that diamond burnishing using an indenter with a working part radius of *R* = 2 mm at a force of *F_b_* = 200 N, an indenter sliding speed of *v_s_* = 50 m/min, a feed rate of *f* = 0.01 mm/rev, and a number of working passes of *n_p_* = 5 increases the microhardness from 225 HV_0.05_ to 400–450 HV_0.05_ by forming a submicronic and nanocrystalline structure. The arithmetical mean deviation of the surface profile reached *Ra* = 0.13 μm.

Study [24] uses the Taguchi method to determine that *F_b_* = 150 N, *f* = 0.025 mm/rev, and *n_p_* = 5 is the optimal combination of technological parameters according to the criterion of maximum microhardness of the surface layer in the nanostructuring burnishing of AISI 304 steel. Not only can the submicronic and nanocrystalline friction structures that form in the surface layer during friction machining determine the main tribological properties of metal alloys to a significant extent but they also can have extremely high mechanical property values [25,26].

Altering the structure of AISI 316L steel to a nanocrystalline state with a ~40 nm grain size by rolling increased the wear resistance of the surface layer under both dry and lubricated friction [27].

Study [28] shows that the wear speed of austenitic steels using dry friction with “stylus-disk” sliding is primarily influenced by tangential velocity. Furthermore, the work revealed the difference in wear speeds of two austenitic steels with different chemical compositions: AISI 304 (0.07% C; 18.48% Cr; 9.04% Ni; 1.85% Mn; 0.36% Mo) and AISI 316 (0.08% C; 18.63% Cr; 9.78% Ni; 1.80% Mn; 2.04% Mo). AISI 316 steel is characterized by higher wear speeds. The nanocrystalline structure of austenitic steels AISI 304 and 316L remains thermally stable during short-term heating up to 600 °C [25,29].

This study aims to establish the influence exerted by the sliding speed and number of passes of a tool’s diamond spherical indenter in the nanostructuring burnishing of AISI 304 steel on the formation of a nanocrystalline structure and modification of the microhardness, texture, and tribological properties of the surface using dry and lubricated friction.

## 2. Materials and Methods

We used disks with thicknesses of 25 mm made of commercial AISI 304 steel hot-rolled rods with diameters of 100 mm as workpieces for our study. The chemical composition of the AISI 304 steel used was C 0.07%; Si 1.0%; Mn 2.0%; Ni 9.555%; S 0.2%; P 0.045%; and Cr 17.824%. The workpieces were subjected to hardening heat treatment to level their structure at 1050 °C for one hour followed by water cooling. The hardness of the material after hardening was 167 HB.

After the heat treatment on a Takisawa EX 300 turn-mill center (Takisawa Machine Tool Co., Okayama, Japan), we proceeded to the mechanical machining of the flat surfaces of the samples, which consisted of preliminary turning followed by nanostructuring burnishing. Preliminary turning was used for the machining of the entire flat surface of the sample. This was followed by the formation of ring areas on the surfaces of the samples using a burnishing tool with a natural diamond. During this process, we varied different machining parameters, including burnishing speed and number of working tool passes. Figure 1a illustrates the nanostructuring burnishing of the sample fixed in the spindle of the machine

The technological parameters of processing are as follows: Preliminary finish turning was performed using a Sandvik WNGA carbide blade with a tip radius of 0.8 mm, a cutting depth of 0.1 mm, a speed of 140 m/min, and a feed rate of 0.15 mm/rev. Subsequent nanostructuring burnishing was carried out using a tool with a natural diamond indenter with a radius of *R* = 2 mm using a Rhenus coolant (Rhenus Lub, Mönchengladbach, Germany), a normal force of 175 N, and a feed rate of 0.025 mm/rev. The variation of the tool’s sliding speed was set in the range of 40–280 m/min with an interval of 80 m/min. Machining was realized in 1, 3, and 5 passes.

After the nanostructuring burnishing of ring areas, we cut out samples to produce angle laps for scanning electron microscopy and foils for transmission electron microscopy using an AgieCut Spirit 20 electrical erosion machine (GF Machining Solutions, Losone, Switzerland). Figure 1b presents a scheme of burnished ring areas of the surface prepared for this study.

The microstructure analysis of the samples’ surface layers after nanostructuring burnishing was carried out using scanning surface layer microscopy and transmission surface layer microscopy with Zeiss Auriga Crossbeam (Carl Zeiss Group, Oberkochen, Germany) and JEOL JEM 2100 (JEOL, Tokyo, Japan) instruments; microdurometry and optical profilometry of the modified surface was performed using an EcoHARD FM300 microhardness tester (Future-Tech Corp., Schwaikheim, Germany) and a Wyko NT-1100 3D profiler (VEECO, Plainview, NY, USA).

To determine the influence of the sliding speed *v*_s_ and the number of tool passes, *n_p_*, on the formation of the microstructure and the size of the nanocrystallites, as well as the microhardness and thickness of the formed nanostructured layer, we conducted experimental studies of nanostructuring burnishing of the workpieces’ surfaces, on which we formed tracks at indenter sliding speeds of *v*_s_ = 40, 120, 200, and 280 m/min with 1, 3, and 5 tool passes.

After the cut-out of the “angle lap with 19°10′ slope angle” samples, they were subjected to abrasing using a scotch pad with grit of P400 to P2000 and polishing with a pad using colloidal silicon. The structures and thicknesses of the modified surface layers of the prepared samples were analyzed using a scanning electron microscope; we also measured the microhardness by sample depth with an interval of ~3.3 μm at a depth of up to 170 μm. To analyze the microstructure using transmission electron microscopy, we prepared foils with thicknesses of 0.3 mm, which were subjected to one-sided abrasing using a scotch pad with a grit of P 200 to a thickness of 200 μm and electropolishing in orthophosphoric acid.

SHIMADZU XRD-7000 (Shimadzu, Tokyo, Japan) and Bruker D8 Advance (Bruker AXS, Karlsruhe, Germany) diffractometers were used for an X-ray structure analysis.

An X-ray structure analysis was carried out using a SHIMADZU XRD-7000 diffractometer with CrK_α_ radiation; the voltage of the tube was 30 kV, and the current of the tube was 30 mA. The analysis determined the phase composition, integral widths of B X-ray lines (111)γ and (110)α, and residual stresses of the surface layers. An X-ray diffraction analysis (XRD) was conducted in the angle range of 2θ = 45–115°; continuous shooting was implemented with a scanning speed of 1 °/min, a pitch of 0.05°, and an exposure time of 3 s. The phase composition was determined using the homologous pairs method. Residual stresses, σ, were determined using the tilt-shift shooting method with lines (220)γ and (211)α.

In addition, an X-ray structure analysis was carried out with a Bruker D8 Advance diffractometer and anode Co (tube voltage 30 kV, tube current 40 mA). An X-ray diffraction analysis (XRD) was conducted in the angle range of 2θ = 45–115°; continuous shooting was implemented with a scanning speed of 0.2°/min, a pitch of 0.05° and an exposure time of 6 s. The phase composition was determined using the Rietveld method with the DIFFRAC.TOPAS V5.0 software. It is known that the Rietveld method demonstrates higher degrees of accuracy than the homologous pairs method, but the former is more labor-intensive.

Tribological properties of the nanostructured surface layer were analyzed using a CETR UMT-3MT tribometer (CETR, Santa Barbara, CA, USA). The counter body was an AISI 440C steel ball (C 0.95–1.2%; Si < 1.0%; Mn < 1.0%; Ni < 1.0%; Cr 16.0–18.0%; Mo < 0.75%), which was 6.3 mm in diameter, with a hardness of 62 HRC. The results of the experiment were processed using Matlab R2023a, MountainMap 9.2, Wyko Vision v. 3.60, and Gwyddion 2.67 software.

## 3. Experimental Results and Discussion

### 3.1. Microhardness and Thickness of the Hardened Surface Layer

Figure 2 shows the variation of microhardness by surface layer depth depending on the number of passes and the tool indenter sliding speed, as measured using the microdurometry of angle lap surfaces. As shown in Figure 2c, at a microhardness of 400 HV, the maximum hardening depth was achieved using five tool indenter passes at a speed of 40 m/min.

An analysis of the obtained dependencies of microhardness HV_0.05_ provides the following conclusions and patterns of surface layer hardening:Preliminary finish turning has a significant impact on increasing the microhardness of AISI 304 steel at a depth of approximately 30 μm, reaching its maximum value of HV_0.05_ in the thin layer, with a thickness of 10 μm;The hardening of the surface layer to the greatest extent at a depth of 150 μm occurs after five tool passes at the sliding speeds of 40 and 200 m/min and reaches about 300 HV_0.05_;A surface microhardness of 350 HV_0.05_ is achieved when using indenter sliding speeds of 40 m/min and 200 m/min at a depth of 60 μm after one and three passes, respectively;After five tool passes at a speed of 40 m/min, a microhardness of 350 HV_0.05_ is found at a depth of 95 μm;A microhardness of HV_0.05_ of the surface layer at an indenter sliding speed of 120 m/min is significantly lower than that at the speeds of 40 m/min and 200 m/min with one and five passes. This may be due to the size of the formed nanocrystallites and the influence of the variation of the structural phase state of AISI 304 steel during nanostructuring burnishing.

A visual analysis of SEM images of the surface layer allowed us to establish that one pass of the indenter yields a thickness of the nanostructured layer of 9 μm at sliding speeds of 40 and 200 m/min (Figure 3a,b). Raising the number of passes to five practically doubles the thickness of the layer with a nanocrystalline structure; the thickness reaches up to 17.7–20.7 μm when the indenter moves at sliding speeds of 200 and 40 m/min (Figure 3c,d).

### 3.2. Determination of Sizes and Calculation of Size Fractions of Nanocrystallites

In study [30], we demonstrate that during the movement of the indenter, the material of the surface layer is characterized by a complex, sign-changing, tense-deformed state. Namely, before the indenter machining, the material exhibits compressed stresses, whereas, after the machining, the stresses change their sign, i.e., an area of tensile stresses is created. At bigger depths, it is manifested by the sign change in shear stresses. Since the indenter made multiple passes on the surface of the machined part, the material of the layer adjacent to the surface was subjected to cyclic sign-changing load, thus creating conditions for the occurrence of the rotational deformation mechanism, which includes both translation and rotation deformation modes. This caused a fragmentation of the original large austenitic grains, followed by the rotation of the resulting fragments. This, in turn, formed narrow high-angle boundaries generated by partial declinations and reduced the crystallites to nanosizes. Microelectronograms that have the form of uninterrupted Debye rings indicate the disorientation of individual fragments to angles of up to a few tens of degrees.

Figure 4 illustrates the distribution of nanocrystallite sizes in the AISI 304 steel surface layer at sliding speeds of 40, 120, 200, and 280 m/min after one, three, and five passes of a sliding indenter; the graphs were drawn based on the analysis of foil TEM images. Grain sizes were determined by measuring the areas of separate elements in dark field images of the structure in the reflection of the α-phase. If it is assumed that all grains have a spherical shape, their sizes, *δ*, can be determined as follows:(1)$$\delta =\sqrt{\frac{4}{\pi }S_{\delta }},$$
where *S_δ_* is the corrected area of the grain’s cross-section.

The spherical shape assumption is based on the rotation shear mechanism of the dispersion of the grain structure by friction under compression. The size fractions of grains were determined using the SIAMS 700 4.1 software package. For each machining mode presented in Figure 4, we produced five samples from one rod of steel. The samples were subjected to heat treatment simultaneously within one batch. To analyze the cross-section areas of grains and understand the allocation of sizes of nanocrystallites, we produced two foils for transmission electron microscopy from each sample. Altogether, the allocation of grain sizes in each machining mode was calculated based on the analysis of cross-section areas of more than 1000 grains.

The corrected area of the grain’s cross-section was calculated as *S_δ_* = *k_s_S_h_*, where *S_h_* is the grain cross-section area measured using the image, and *k_s_* is a correction factor that factors in the random dissection of the grain during the manufacture of the foil. Based on the solution of a geometrical problem of the grain dissection in a random spot, the correction factor *k_s_* = 1.5 was determined as the mathematical expectation of the ratio of the arbitrary cross-section area to the half-section area.

It was established that more than 80% of grains had sizes smaller than 100 nm and high angular disorientation after three indenter passes at a tool indenter sliding speed of 40 m/min (Figure 4b).

In comparison, one tool pass at indenter sliding speeds from 40 to 280 m/min (Figure 4a,d,g,j) and five tool passes at speeds of 400 and 200 m/min (Figure 4c,l) yielded a mixed structure that contained less than 15% of nanocrystallites with sizes greater than 100 nm.

The results of transmission electron microscopy allowed us to conclude that the average size of the formed nanocrystallites increases with the increase in the tool indenter sliding speed. The minimum nanocrystallite size (53 to 92 nm) was obtained after three consecutive tool passes with a tool sliding speed from 40 to 280 m/min (Figure 5). Thus, when forming nanostructures on the surface layer of AISI 304 steel with one, three, and five passes, the optimal tool sliding speed is 40 m/min.

Figure 6 shows bright field images and microdiffractions of the microstructure of the nanocrystallized surface layer after three indenter passes *n_p_* = 3 at sliding speeds of 40 m/min, 200 m/min, and 280 m/min. Bright field images and microdiffractions of the AISI 304 steel structure after nanostructuring burnishing with three passes at a sliding speed of 280 m/min demonstrate the high productivity and effectiveness of this nanocrystallization method for *n_p_* = 3.

### 3.3. Change in Phase Composition and Residual Stresses by Depth of the Surface Layer

After finish turning and nanostructuring burnishing, we determined the content of residual austenite γ and the integral breadth of X-ray lines (111))γ and (110)α, reflecting the location density and microdistortions in austenite and deformation of martensite and δ-ferrite, which are given in Table 1. This study was carried out using a SHIMADZU XRD-7000 diffractometer. Figure 7 shows the allocation of residual austenite γ content values by the depth of the surface layer, as obtained when consequently electrolytically removing the metal from the part’s surface. Table 1 and Figure 7 suggest that up to 90 vol.% of austenite is retained immediately on the part’s surface and in the layer with a depth of 50 μm after finishing turning with a carbide blade, while the content of austenite in the initial structure of the quenched steel was not less than 95 vol.%.

Hence, turning yielded only 5 vol.% of deformation martensite in the surface layer of a sample made of austenitic steel. This heightened stability of AISI 304 steel austenite against deformation phase transformation under finish turning may be connected with the use of a Viper-type tool plate with a large tip radius of 1.2 mm.

Data presented in Table 1 indicate that one-pass burnishing of a part after a preliminary finish turning does not result in a change in the phase composition of the surface. After burnishing with one pass (*n_p_* = 1), the austenite fraction was 90 vol.%, which was the same as after turning. After burnishing with *n_p_* = 3 passes, the content of austenite on the part’s surface decreases to 40 vol.%, indicating the deformation transformation of a significant fraction of austenite (~50 vol.%) into deformation martensite. Burnishing with *n_p_* = 5 passes results in an almost complete deformation decomposition (~85 vol.%) of austenite; not more than 5 vol.% of γ-phase was registered on the part’s surface.

Figure 7 illustrates that the content of austenite decreases with the depth from the surface of the part machined by burnishing, with the number of indenter passes of *n_p_* = 1 and *n_p_* = 3 reaching the minimum values of 55 and 8 vol.% of γ-phase, respectively, at a depth of *h* ~40 μm. With further distance from the surface, however, we observed the continuous growth of austenite content, amounting to up to 90 vol.% at a depth of *h* = 150 μm for *n_p_* = 1 and *n_p_* = 3. Nonmonotonous (with minimum) variation of the phase composition by the parts’ surface layers after nanostructuring burnishing with *n_p_* = 1 and *n_p_* = 3 indenter passes is due to the simultaneous influence of two factors that produce opposite effects on the degree of deformation decomposition of austenite.

On the one hand, the stabilization of austenite to deformation γ→α′-martensite transformation decreases with the depth from the turning surface because the heating and speed of the deformation leading to this stabilization are at their maximum values immediately on the part’s surface and decrease with the depth of the surface layer. This is the dominating factor in the layer with a thickness of ~40 μm, providing for the observed decrease in the fraction of austenite and, consecutively, an increase in the intensity of its deformation decomposition. On the other hand, the degree of metal deformation decreases with depth from the surface. This results in a less effective deformation decomposition of austenite and a corresponding growth of γ-phase content at a depth of 40 μm from the part’s surface.

After nanostructuring burnishing with five tool passes, the minimum content (≤5 vol.%) of γ-phase corresponds to an almost complete deformation decomposition of austenite in the layer at a depth of up to 100 μm (Figure 7, *n_p_* = 5). At a depth from the burnished surface starting from more than 100 μm (up to 400 μm), the content of austenite continuously increases, reflecting the less intensive deformation γ→α′-martensite transformation as the degree of deformation of the metal decreases. Therefore, the examined results of the X-ray analysis indicate a very high degree of dependency on the phase composition of the surface layer of a part made of austenitic steel from the modes of turning and burnishing on the turn-mill center. Thus, the deformation conditions of the part’s surface in finish turning by a carbide blade with a tip radius of 1.2 mm result in an almost complete stabilization of austenite to the deformation γ→α′-transformation in the depth of the surface layer.

One-pass burnishing decreases the content of austenite at a depth of ~40 μm from the surface to 55 vol.% (Figure 7, *n_p_* = 1). A higher degree of accumulated deformation obtained by burnishing with *n_p_* = 3 indenter passes decreases the austenite content to 40% vol.% on the part’s surface and to 8–10 vol.%—at a depth of 30–50 μm (Figure 6, *n_p_* = 3). Multi-pass burnishing with *n_p_* = 5 indenter passes results in a practically complete deformation decomposition of austenite, with 85…90 vol.% of deformation martensite forming on the part’s surface layer, with a thickness of 100 μm.

Results presented in Table 1 also demonstrate that finish turning and burnishing with *n_p_* = 1 and *n_p_* = 3 passes causes the integral breadth of the X-ray line (111)γ to rise from 10 min to 36–43 min. This reflects the increase in location density in austenite and microdistortions of its face-centered cubic lattice. The maximum integral breadth of the X-ray line of the α-phase (B_(110)α_ = 59 min) was discovered on the steel surface after finish turning, when there was 10 vol.% of α-phase consisting of deformation martensite and δ-ferrite deformed by turning. After nanostructuring burnishing with a varying number of indenter passes, the breadth of the line of the α-phase has the values of B_(110)α_ = 33–37 min.

Plots in Figure 8 demonstrate that a part machined by turning and burnishing has the maximum integral breadth of lines of the γ- and α-phases on the surface layer, with a thickness of a few micrometers. Generally, the levels of X-ray characteristics under investigation (B_(111)γ_ and B_(110)α_) decrease with distance from the burnished surface, indicating a decrease in the number of defects and microdistortions of the crystal structure of analyzed phases.

Data presented in Table 2 and Figure 9 characterize the deformation hardening of the surface of austenitic steel during finish turning and burnishing in different modes, as well as the allocation of microhardness by the depth of surface layers of parts machined on a turn-and-mill center.

It was established that turning already increases the microhardness of the part’s surface from HV_0.05_ = 167 in its initial tempered state to 300 HV_0.05_. Consequent burnishing with *n_p_* = 1, *n_p_* = 3, and *n_p_* = 5 indenter passes causes further hardening of the surface to 400–480 HV_0.05_ (Table 2). It was also established that intensive deformation hardening of austenitic steel occurs due to the formation of fragmented submicrocrystalline and nanocrystalline austenitic, martensitic–austenitic, and martensitic structures with an elevated number of defects (dislocation density) and microdistortions of the crystal structures of the γ- and α-phases (Figure 8) as a result of the joint effect of finish turning and burnishing.

It is important to note that even though there was as much as 90 vol.% of austenite on the surface of the steel under analysis (Table 1), its microhardness was relatively high, at 300 HV_0.05_ and 400 HV_0.05_ (Table 2).

The microhardness of the thin surface layer, which consists primarily of a nanocrystalline structure with an average grain size of 57 nm, reaches 460 HV_0.05_ after three indenter passes (Table 2). After five indenter passes, a microhardness of 400 HV_0.05_ is retained in the layer up to 90 μm in depth (Figure 2c).

We analyzed the influence of turning and burnishing on the stressed state of the surface layer of a part made of AISI 304 steel with different numbers of indenter passes. Residual stresses in the γ-phase (σ_γ_) and α-phase (σ_α_) were determined using an angled X-ray of lines (200)γ and (211)α.

An analysis of the part’s surface after finish turning revealed minor residual compressing stresses σ_γ_ = −60 MPa. However, finish turning causes positive residual stresses (σ_α_ = +550 MPa tensile stresses) immediately on the part’s surface; these stresses abruptly decrease with the depth from the surface of the deformed layer and transfer to compressing residual stresses at a maximum level of σ_α_ = −800 MPa at a depth of ~20 μm (Figure 9a, plot T). We observed that with distance from the turned surface, the compressing stress level decreased substantially to σ_α_ = −250…−20 MPa.

Undesired significant compression stresses on the turned surface in the α-phase and the high inhomogeneity of the stressed state in the surface layer of the turned part (Figure 9a, plot T) may be due to intensive destruction and local heating of the surface, as well as the non-uniform distribution of temperatures and deformations by depth of the surface layer during turning.

Data from Table 2 suggest that nanostructuring burnishing with different numbers of passes transforms tensile residual stresses in the α-phase on the surface of the turned surface into compressing stresses σ_α_ from −610 to −380 MPa. Simultaneously, the maximum level of compressing stresses in the α-phase on the part’s surface (σ_α_ = −610 MPa) was discovered after a one-pass burnishing, which did not change the phase composition of the turned surface (fraction of α-phase was 10 vol.%). After three and five indenter passes, when the amount of martensite on the part’s surface increased significantly (see Table 1), residual stresses in the α-phase were σ_α_ = −470 and σ_α_ = −380 MPa, respectively (Table 2).

According to Figure 9, burnishing in different modes stabilizes the finish turning-induced heterogeneous stressed state of the part’s surface layer; after burnishing, the surface layer was characterized by the presence of compressing residual stresses in the α-phase σ_α_ = −700…−250 MPa.

Table 2 also demonstrates that burnishing with *n_p_* = 1 and *n_p_* = 3 indenter passes results in stresses that are relatively higher than the favorable level of compressing stresses in the γ-phase on the turned surface (from σ_γ_ = −60 MPa to σ_γ_ = −90 MPa).

Since burnishing with a different number of passes fundamentally changes the phase composition both on the surface (see Table 1) and in the depth of the surface layer of a part made of AISI 304 steel, the contribution of residual stresses in the γ- and α-phases to the stressed state of the surface layers will depend on the number of burnishing passes. Specifically, after three indenter passes, when the fraction of the α-phase reaches 60% vol.% on the part’s surface and 90–92 vol.% at a depth of 30–50 μm (Figure 7, plot *n_p_* = 3), the contribution of residual stresses σ_α_ to the stressed state of the surface layer must be higher than that occurring in the case of one-pass burnishing, which is characterized by higher contents of austenite in the structure of the part’s surface layer (Figure 7, plot *n_p_* = 1). The stressed state of the surface layer after burnishing with *n_p_* = 5 indenter passes will be determined by residual stresses in the α-phase to an even higher extent.

Ergo, utilizing the analyzed number of tool passes during the nanostructuring burnishing of AISI 304 austenitic stainless steel, which is subjected to preliminary turning, yields close levels of microhardness in the thin surface layer with a thickness of 40 μm: 400–480 HV_0.05_ (Table 2). It was discovered that variation in the number of passes of the burnishing tool may be used to change the phase composition (see Table 1, Figure 7), effective hardening depth, and stressed state of the surface layer (Table 2, Figure 9).

In order to confirm the obtained surface layer phase composition data, we subjected an additional sample to nanostructuring burnishing in the optimal mode that formed a nanocrystalline structure with the minimum grain size (indenter sliding speed 40 m/min, feed rate 0.025 mm/rev, force 175 N) in one, three, and five indenter working passes. A Bruker D8 Advanced diffractometer was used for the tests. The phase composition determined using the Rietveld method is presented in Figure 10 and Table 3.

A comparison of diffractograms demonstrates that the (111)γ, (200)γ, (220)γ, and (311)γ peak intensities have the maximum value in the original state and decrease when more passes are added. After the first nanostructuring burnishing pass, martensite peaks (110) + (011)α, (200) + (020)α, and (211) + (112)α emerge, and their intensity increases as more passes are added. On the diffractogram, a martensite doublet is not visible due to the low degree of tetragonality.

### 3.4. Tribological Properties of the Hardened Surface Layer

To understand the influence of the parameters of surface topography of hardened areas on tribological properties with dry and lubricated friction, we studied their formation when using the indenter sliding speed that is optimal for nanocrystallization: *v*_s_ = 40 m/min. We analyzed the arithmetical mean height (*Sa*), void volume (*Vvv*), and reduced peak height (*Spk*) parameters, as they determine the wear resistance of the surface to the greatest extent. Variation of these topography parameters depending on the number of tool passes is presented in Figure 11. Based on the presented research results, we can predict that in tribological tests using dry friction, the lowest coefficient of friction (COF) will be exhibited by the surface obtained with one nanostructuring burnishing pass, since the *Sa* and *Spk* parameters have the lowest values of 0.043 μm and 0.049 μm, respectively. Using lubricated friction, the lowest friction coefficient will be exhibited by a surface obtained after five passes because the *Vvv* parameter will have the highest value of 1.97 × 10^−5^ mm^3^/mm^2^.

The tribological tests used the ball-on-disk scheme and were carried out on a CETR UMT-3 (CETR, USA) machine. We produced disk-shaped workpieces with thicknesses of 25 mm and diameters of 100 mm as samples for tribological tests. To install the workpieces in the tribological machine, holes with diameters of 6.5 mm were drilled in their centers. On the surfaces of each sample, we created three concentric areas with widths of 10 mm that were machined using nanostructuring burnishing with one, three, and five passes at an optimal sliding speed of *v_s_* = 40 m/min and force of *F_b_* = 175 N. As demonstrated in Figure 12a, the concentric area situated closer to the outer edge of the workpiece, the middle area, and the inner area were subjected to nanostructuring burnishing with one, three, and five passes, respectively.

A ball with a diameter of 6.3 mm and hardness of RC 62 made of AISI 440-C steel was used as the counter body in the tribological tests. During the experiments, the counter body moved within the concentric areas with a velocity of 2 mm/s for 1800 s (30 min) and a normal load of 2 N.

For tests under dry friction conditions, the sample was fixed immediately on the rotor of the tribological machine (Figure 12b). During tribological tests using dry friction, the ball-shaped counter body moved along the surface of the workpiece with a circular trajectory with radii of r_1_ = 42, r_2_ = 29, and r_3_ = 14 mm at one, three, and five passes, respectively. It formed wear tracks that are presented in Figure 11a.

Under lubricated conditions, the sample was submerged into a bath with SINTEC TM4 SAE 75W-90 transmission oil at 25 °C, and the counter body movement radii were r_1_ = 35, r_2_ = 25, and r_3_ = 13 mm with one, three, and five passes, respectively. The counter body load on the surface was 2 N. The linear speed was 2 mm/s.

The coefficient of friction (COF) analysis results are presented in Figure 13. Using dry friction, after breaking in at a distance of 500 mm, the minimum friction coefficient was exhibited by the surface obtained after one tool pass (Figure 13a), corresponding to the previous prognosis. Using lubricated friction (Figure 13b), the minimum friction coefficient values after grinding were obtained on surfaces machined by three and five indenter passes. Thus, this experiment confirmed the forecast presented above.

The wear intensity of the AISI 304 steel surface after nanostructuring burnishing was assessed by analyzing the width and profile of the formed friction tracks (Figure 14). The obtained width of the friction tracks suggested that dry friction leads to higher wear than that yielded by lubricated friction. Tracks formed using dry friction exhibit traces of adhesive bonding. To correctly determine the parameters of friction tracks, we cleaned the workpiece of oil and wear products in an ultrasound bath with dimethyl sulfoxide, oscillations at a frequency of 60 kHz, and a temperature of 80 °C.

The morphology of friction tracks formed during tribological tests is characterized by traces of microwelding and microbonding of contacting surfaces, as well as traces of localized material tear-off. Friction tracks formed after testing had widths of 65.93–67.52 μm using dry friction and 49.61–57.94 μm using lubricated friction.

Figure 15 presents a profilogram of the cross-section of a friction track after five tool passes, lubricated friction tests, and the removal of wear products.

Figure 16 presents optical microscopy (×500) of areas of friction tracks after tribological tests of nanostructured surfaces formed using dry friction and lubricated friction after one and five tool passes.

The wear track formed using dry friction on the surface machined by nanostructuring burnishing with one tool pass has prominent traces of adhesion bonding manifested as formed micropits with sizes of 3 to 8 μm (Figure 16a). When there is counter body friction on an analogous surface under lubricated conditions, the wear track is dominated only by the extrusion of material on the edge of the track (Figure 16b). The wear track obtained on a nanostructured surface after five tool passes under dry friction conditions exhibits multiple traces of material tear-off in the form of pits, as well as traces of adhesive bonding (Figure 16c). A wear analysis of the surface formed after five burnishing tool passes under lubricated conditions revealed no traces of adhesive bonding. There were solely individual micro tear-offs and an elevated degree of extrusion on the edge of the track (Figure 16d)

The specific wear coefficient of the surface was calculated using the classic Archard formula considering the set friction path, normal counter body load, and calculated area of the cross-section of the friction track, as follows [31]:(2)$$k=\frac{V_{\mu }}{Fl_{\mu }},$$
where *V_µ_* is the volume of the worn material, *F* is the normal ball load, and *l_µ_* is the length of the friction track.

The volume of the worn material *V_µ_* was calculated using the following formula:(3)$$V_{\mu }=2\pi RS_{\mu },$$
where *R* is the radius of the friction track in relation to the center of the sample, and *S_µ_* is the cross-section area of the friction track.

The cross-section area of the friction track was calculated as the geometric area of a section of the spherical contact that had been flattened due to wear, as follows:(4)$$S_{\mu }=r^{2}\arcsin\left(\frac{w_{\mu }}{2r}\right)-\frac{w_{\mu }}{4}\sqrt{4r^{2}-w_{\mu }^{2}},$$
where *r* is the counter body radius, and *w_µ_* is the friction track width.

The established parameters of friction tracks were used to build bar charts reflecting the calculated volume of worn material and specific wear coefficient for one, three, and five indenter passes (Figure 17). The radii of friction tracks formed on the disk with nanostructured surfaces after machining by burnishing were *r*_1_ = 42 mm (*n*_p_ = 1), *r*_2_ = 29 mm (*n*_p_ = 3), and и *r*_3_ = 14 mm (*n*_p_ = 5) (Figure 12a). In tribological tests under lubricated conditions, friction tracks were formed on hardened surfaces on the reverse side of the disk with similar radii from the center and a friction distance of 3.5 m.

Based on the data on the volume of worn material and specific wear coefficient, it can be concluded that there is a correlation between wear parameters and the obtained microhardness of the surface (Figure 2).

The minimum values of the volume of worn material and specific wear coefficient are reached with five passes of the burnishing tool for both dry and lubricated friction.

## 4. Conclusions

The obtained microhardness of AISI 304 steel surfaces machined using nanostructuring burnishing was 480 HV_0.05_ and is comparable to that produced by low-temperature plasma carburizing (475 HV_0.05_) in study [10]. The roughness of the burnished surfaces was Sa = 0.04–0.15 μm, which is comparable to that obtained by sandblasting in [11] (Sa = 0.05 μm). The maximum thickness of the nanocrystallized layer was 20.7 μm, and the hardened layer had a microhardness of 400 HV_0.05_ practically to 90 μm after five burnishing tool passes, which is significantly higher than the surface nanocrystallization depths obtained by nitrocarburizing and sandblasting presented in the review. Coefficients of friction measured during “ball-on-disk” studies yielded by both dry and lubricated friction are comparable to the coefficient of friction of a polished surface (0.4 using dry friction and 0.15 using lubricated friction) established in [12] and significantly lower than those produced by other mechanical machining methods. The established specific wear coefficient of the surface with a nanocrystalline structure after nanostructuring burnishing was 5.83 × 10^−15^ m^3^/(N·m), which is an order of magnitude lower than 1.13 × 10^−14^ m^3^/(N·m), which was achieved by sandblasting in [11].

We compared the mechanical and tribological properties of AISI 304 steel that were obtained after nanostructuring burnishing with those of magnetron coatings with CrSiN and WSiN silicides that have high hardnesses of 38–45.4 GPa. However, these coatings have low thicknesses of 2–4.5 µm. Additionally, their friction coefficients are 0.3 and 0.4, while their wear intensities are comparable to nanostructuring burnishing and equal to 2 × 1^−15^ and 5 × 1^−16^ m^3^/(N·m), respectively.

The maximum hardness of 887 HV_0.03_and coating thickness of 200 µm can be obtained by applying iron-containing titanium carbide powder using the HVAF method. Nevertheless, having a relatively high friction coefficient of µ = 0.41 and a wear coefficient amounting to 5 × 10^−14^ m^3^/(N·m), coatings do not exhibit advantages over multi-pass machining using nanostructuring burnishing. Moreover, it should be noted that functional coatings do not provide gradual changes in properties by surface depth. A layer formed by five passes of a sliding indenter exhibits a gradual gradient of hardness change at depths of up to 160 µm and exceeds the microhardness of AISI 304 steel in its original state by 64.5%.

The data in Table 4 represent a comparative overview of various methods of AISI 304 stainless steel surface hardening.

It was discovered that the intensive deformation hardening of austenitic steel occurs due to the formation of fragmented submicrocrystalline and nanocrystalline austenitic, martensitic–austenitic, and martensitic structures, with elevated degrees of deficiency (dislocation density) and microdistortions of crystal structures of the γ- and α-phases, as a result of the joint effects of finish turning and burnishing. One-pass burnishing of AISI 304 steel at the optimal sliding speed of *v_s_* = 40 m/min decreases the content of austenite at a depth of ~40 μm from the surface to 72 vol.%. Burnishing with *n_p_* = 3 indenter passes decreases the amount of austenite to 37 vol.%, while five-pass burnishing causes a quite significant deformation decomposition of austenite, with the formation of 80 vol.% of deformation martensite in the surface layer with a thickness of 100 μm.

Multi-pass nanostructuring burnishing technology enables the user to form nanocrystallites with an average size smaller than 60 nm in the surface layer and obtain a microhardness of 400 HV_0.05_ at a depth of up to 85 μm and a sliding speed of 40 m/min with three indenter passes.

If the indenter sliding speed is raised to 280 m/min, it causes the size of the nanocrystallites to increase up to 92 nm, resulting in the microhardness decreasing to 320 HV_0.05_ at a depth of 60 μm.

Using a higher number of passes with an indenter speed of 40 m/min causes an increase in roughness. Simultaneously, raising the number of passes to five causes the surface texture parameters related to oil absorption capacity (for example, the void volume (*Vvv*) parameter) to increase by up to more than four times. This has a favorable effect on the wear resistance of the surface under oiled friction.

The COF values obtained in “ball-on-disk” tribological tests under dry friction were lower than those presented by other authors and amounted to 0.22 after one and three tool passes and 0.3 after five tool passes after covering a distance of 2 m on the surface. An increase in the distance destabilizes the friction process, causing the COF value to rise to 0.27–0.35 if the path is 3.5 m. Under lubricated conditions, the friction coefficient of surfaces machined using nanostructuring burnishing with three and five tool passes is the same and amounts to 0.15, practically retaining this value until the path is prolonged to 3.5 m.

## Figures and Tables

**Figure 1 materials-17-05656-f001:**
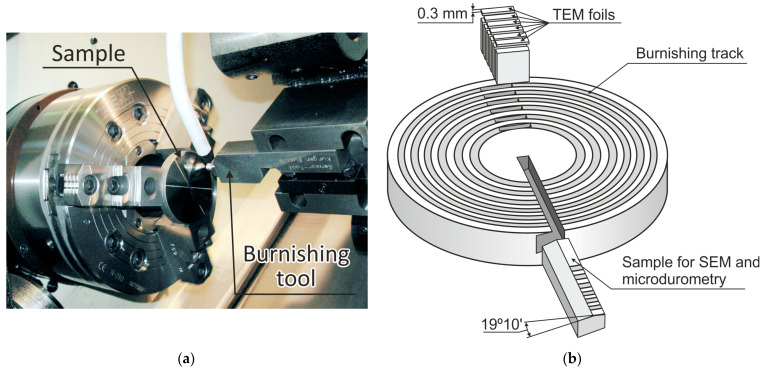
Nanostructuring burnishing on Takisawa EX 300 machining center (Takisawa Machine Tool Co., Japan) (**a**); scheme of cut-out of samples for microdurometry and SEM of angle laps (oblique cross-section) and TEM foils (**b**).

**Figure 2 materials-17-05656-f002:**
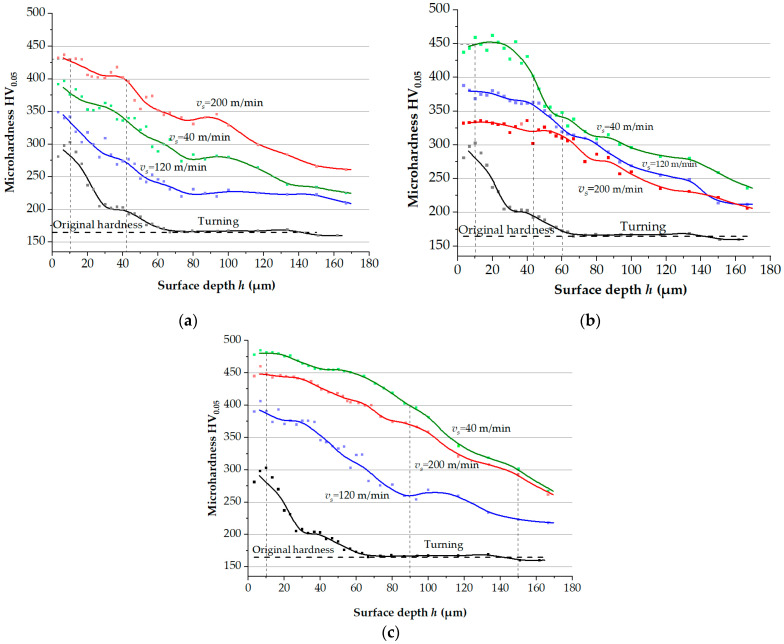
Plots of microhardness variation by surface layer depth after turning and nanostructuring burnishing with one (**a**), three (**b**), and five (**c**) indenter passes.

**Figure 3 materials-17-05656-f003:**
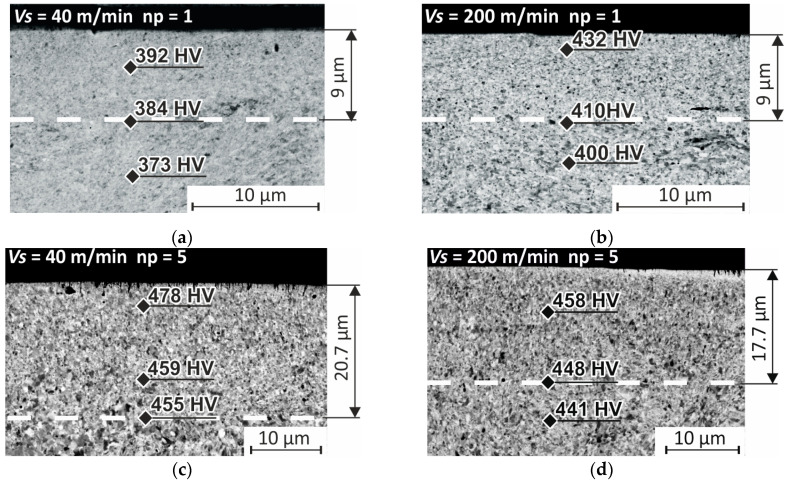
SEM images of the thin surface layer of AISI 304 steel after burnishing: (**a**) *v_s_* = 40 m/min, *n_p_* = 1; (**b**) *v_s_* = 200 m/min, *n_p_* = 1; (**c**) *v_s_* = 40 m/min, *n_p_* = 5; (**d**) *v_s_* = 200 m/min, *n_p_* = 5.

**Figure 4 materials-17-05656-f004:**
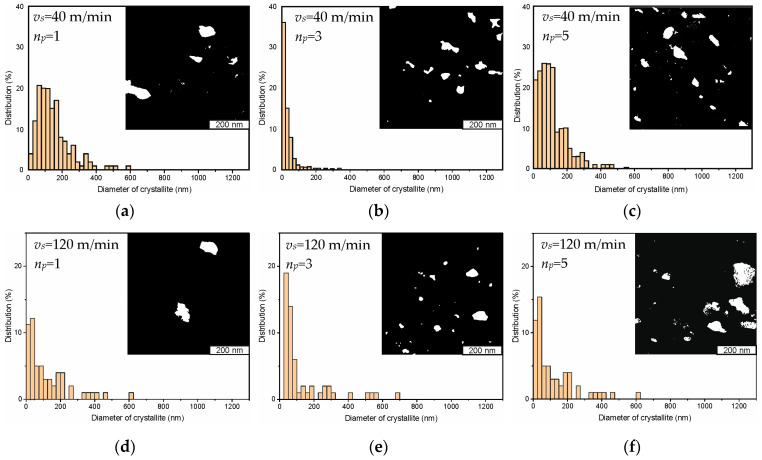
Histograms of the allocation of nanocrystallite sizes on the surface layer of AISI 304 steel at the following sliding speeds: 40 m/min (**a**), 120 m/min (**d**), 200 m/min (**g**), 280 m/min (**j**) after one indenter pass; 40 m/min (**b**), 120 m/min (**e**), 200 m/min (**h**), 280 m/min (**k**) after three indenter passes; and 40 m/min (**c**), 120 m/min (**f**), 200 m/min (**i**), 280 m/min (**l**) after five indenter passes.

**Figure 5 materials-17-05656-f005:**
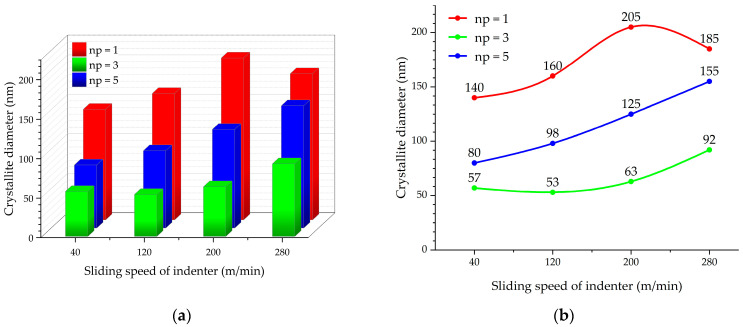
3D (**a**) and 2D (**b**) plots of the influence of the indenter sliding speed on the average grain size with one, three and five consecutive tool passes.

**Figure 6 materials-17-05656-f006:**
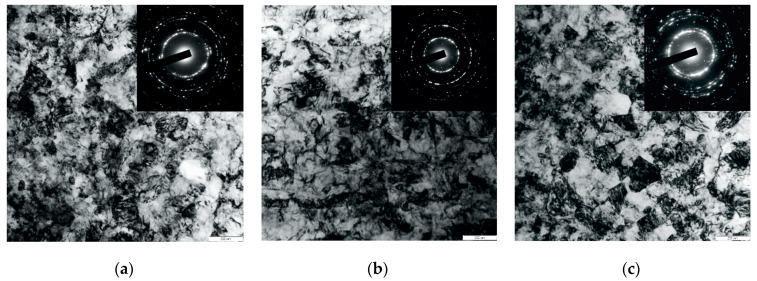
Bright field images and microdiffractions of the microstructure of the nanocrystallized surface layer after three indenter passes *n_p_* = 3 at sliding speeds of 40 m/min (**a**), 200 m/min (**b**), and 280 m/min (**c**).

**Figure 7 materials-17-05656-f007:**
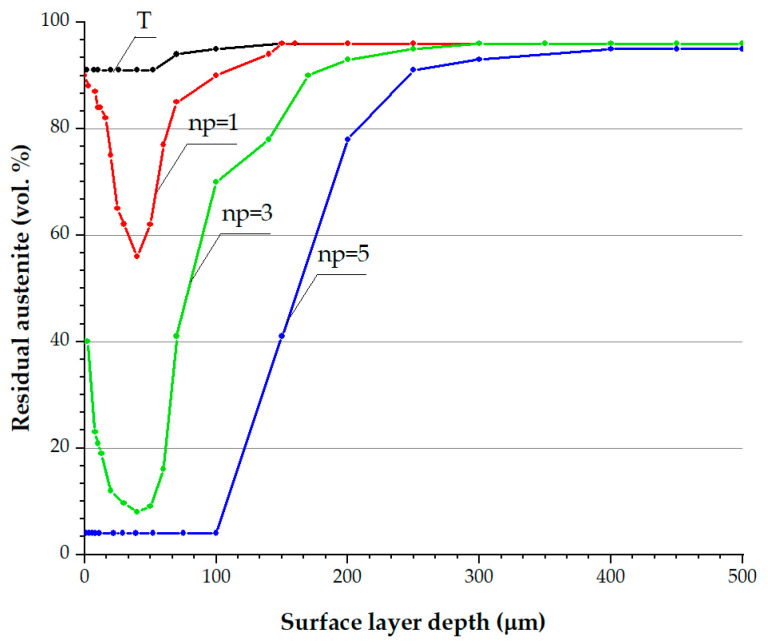
Plots reflecting the variation of residual austenite content (γ) in the structure after turning (T) and nanostructuring burnishing with the number of indenter passes of *n_p_* = 1, *n_p_* = 3, and *n_p_* = 5.

**Figure 8 materials-17-05656-f008:**
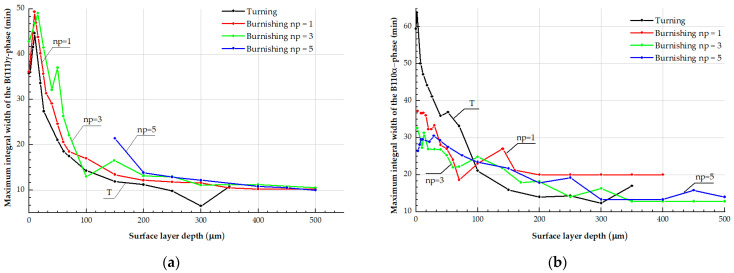
Plots reflecting the variation of the integral breadth of X-ray lines (111)γ (**a**) and (110)α (**b**) by the depth of the surface layer of a part made of austenitic AISI 304 steel after turning (T) and nanostructuring burnishing with the following numbers of indenter passes: *n_p_* = 1, *n_p_* = 3, and *n_p_* = 5.

**Figure 9 materials-17-05656-f009:**
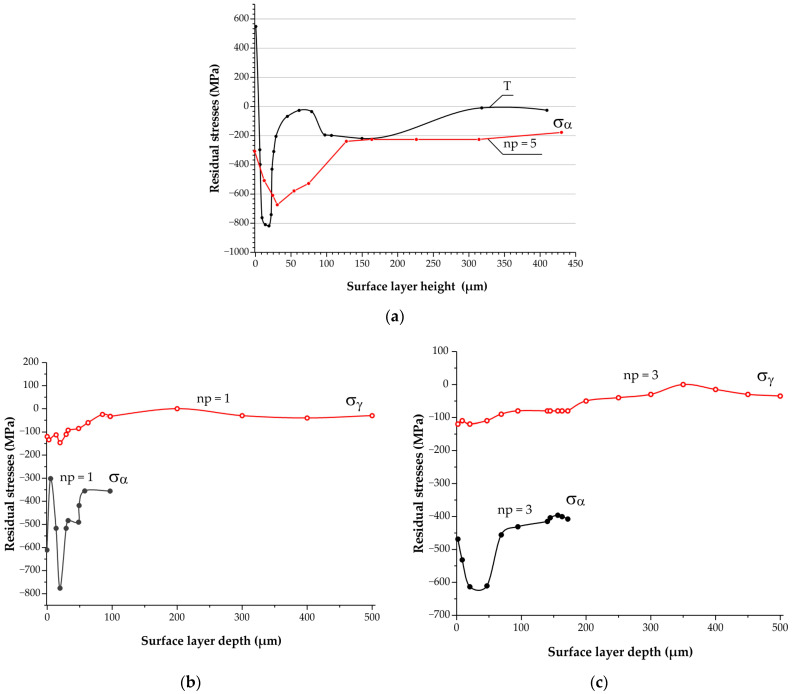
Variation in residual stresses in the α-phase (σ_α_) and γ-phase (σ_γ_) by depth of the AISI 304 steel surface layer after turning (T) and burnishing with *n_p_* = 1 (**b**), *n_p_* = 3 (**c**), and *n_p_* = 5 (**a**) indenter passes.

**Figure 10 materials-17-05656-f010:**
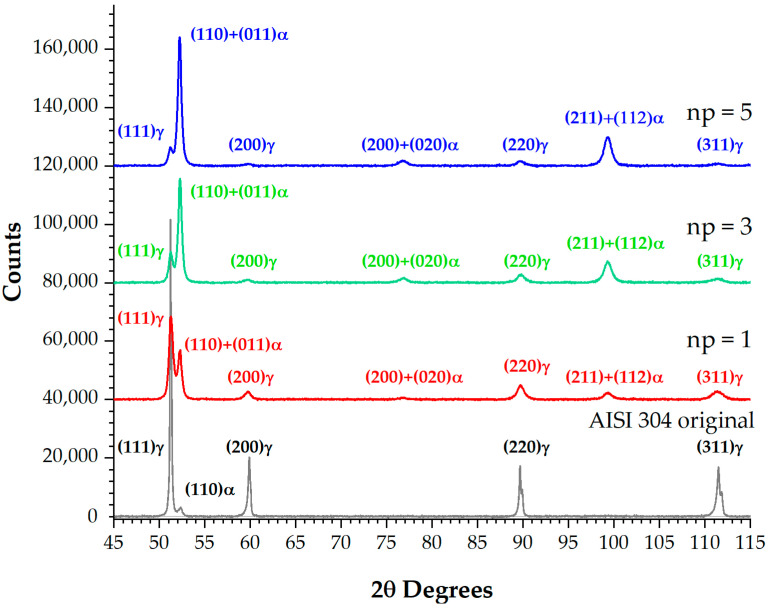
X-ray diffraction patterns of AISI 304 steel samples in the original state and after nanostructuring burnishing with the following numbers of passes: *n_p_* = 1, 3, and 5.

**Figure 11 materials-17-05656-f011:**
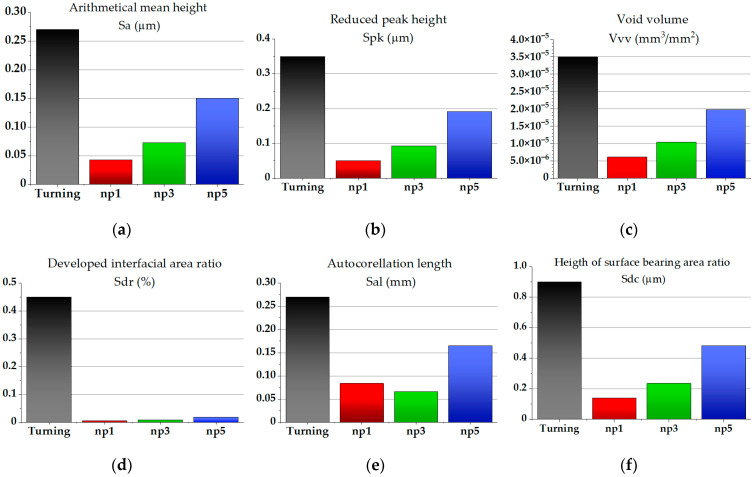
Topography parameters of a disk after finish turning and nanostructuring burnishing of AISI 304 steel with 1, 3, and 5 tool passes with a sliding speed of *v*_s_ = 40 m/min: (**a**) *Sa*; (**b**) *Spk*; (**c**) *Vvv*; (**d**) *Sdr*; (**e**) *Sal*; and (**f**) *Sdc*.

**Figure 12 materials-17-05656-f012:**
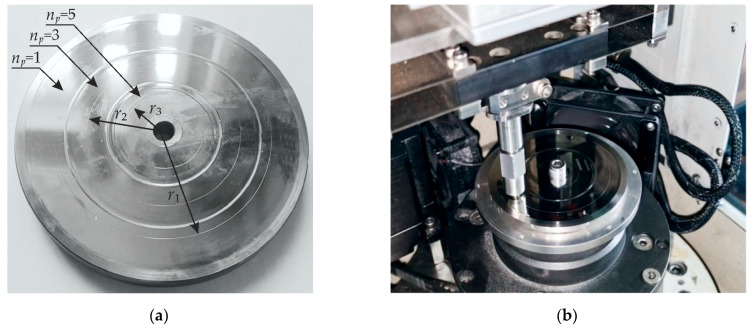
Disk with nanostructured surfaces (**a**) and experimental tests of tribological properties on a CETR UMT-3MT tribometer (CETR, USA) (**b**).

**Figure 13 materials-17-05656-f013:**
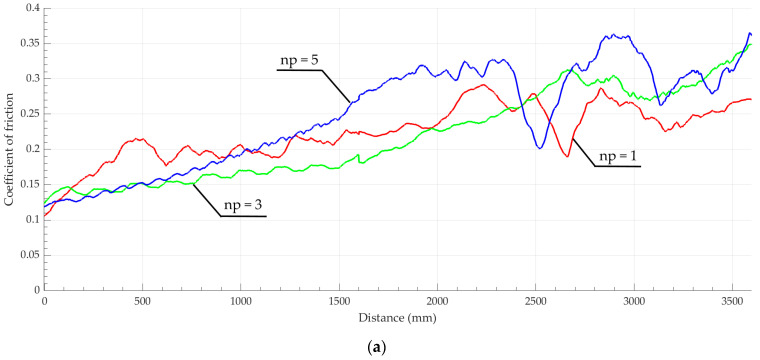
Coefficient of dry friction (**a**) and lubricated friction (**b**) of the AISI 304 steel surface after nanostructuring burnishing when using the “ball-on-disk” scheme at a distance of 3600 mm.

**Figure 14 materials-17-05656-f014:**
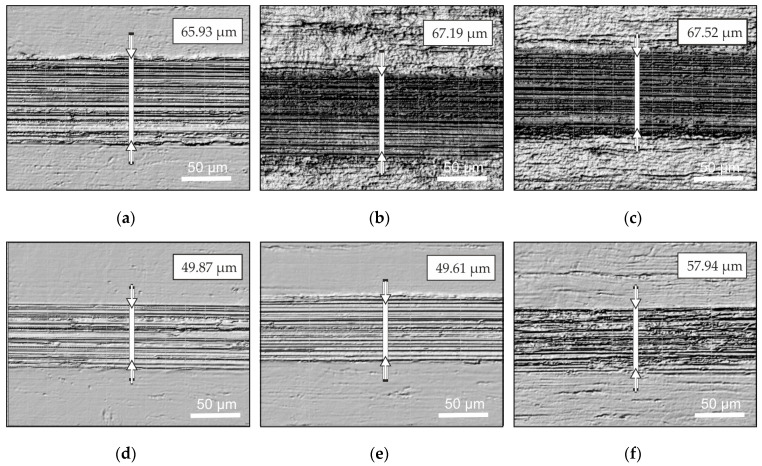
Optical microscopy of friction tracks (×500) formed during tribological tests of nanostructured surfaces after one (**a**), three (**b**), and five (**c**) indenter passes using dry friction and one (**d**), three (**e**), and five (**f**) indenter passes using lubricated friction.

**Figure 15 materials-17-05656-f015:**
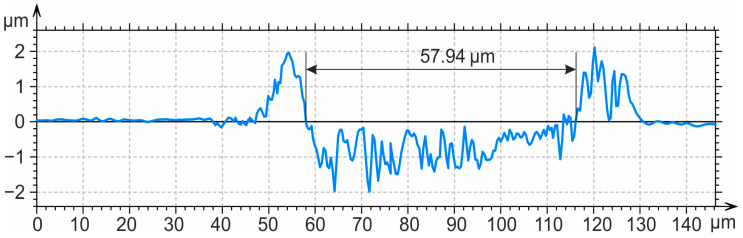
Profilogram of the cross-section of a friction track on the surface after nanostructuring burnishing (*n_p_* = 5) using lubricated friction and the removal of wear products.

**Figure 16 materials-17-05656-f016:**
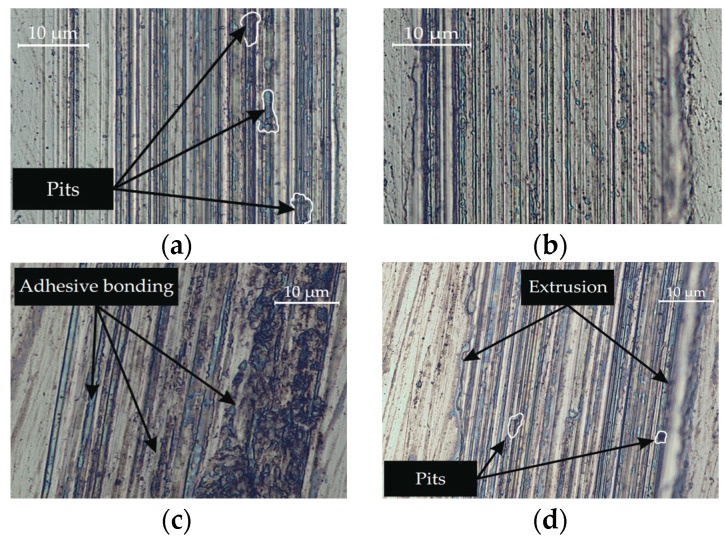
Optical microscopy (×500) of areas of friction tracks after tribological tests of nanostructured surfaces formed using dry friction after one (**a**) and five (**c**) tool passes and lubricated frictionafter one (**b**) and five (**d**) tool passes.

**Figure 17 materials-17-05656-f017:**
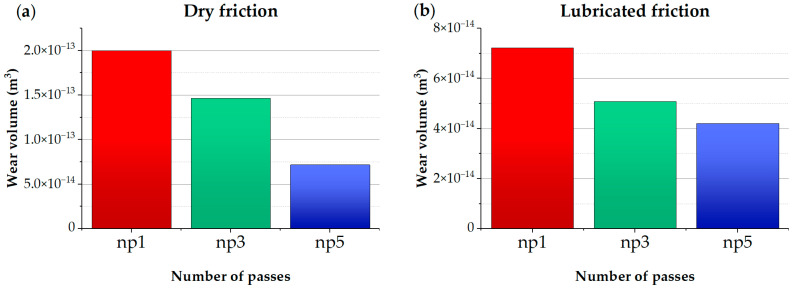
Volume of worn material (**a**,**b**) and specific wear coefficient (**c**,**d**) under dry and lubricated friction conditions.

**Table 1 materials-17-05656-t001:** Influence of finish turning and the number of tool passes *n*_p_ in nanostructuring burnishing on the content of residual austenite γ and the integral breadth of X-ray lines B_(111)γ_ and B_(110)α_.

State of the Sample	*n_p_*	γ, vol. %	B_(111)γ_, min	B_(110)α_, min
Initial	–	≥95	10	–
Finish turning using an Intool WNMG080412 blade with a tip radius of 1.2 mm at a cutting depth of 0.1 mm; speed of 140 m/min; and feed rate of 0.15 mm/rev	–	90	36	59
Burnishing using an indenter with a radius of *R* = 2 mm at a sliding speed of *v_s_* = 40 m/min; burnishing force of *F_b_* = 175 N; feed rate of *f_b_* = 0.025 mm/rev; and number of passes of *n_p_* = 1, 3, 5	1	90	36	33
	3	40	43	34
	5	≤5	–	37

**Table 2 materials-17-05656-t002:** Impact of finish turning and multi-pass nanostructuring burnishing on microhardness HV_0.05_ and residual stresses in the γ-phase (σ_γ_) and α-phase (σ_α_) on the surfaces of parts made of AISI 304 steel.

State of the Sample	*n_p_*	HV_0.05_	σ_γ_, MPa	σ_α_, MPa
Initial	–	167	0	–
Finish turning using an Intool WNMG080412 blade with a tip radius of 1.2 mm at a cutting depth of 0.1 mm; speed of 140 m/min; and feed rate of 0.15 mm/rev	–	300	–60	+550
Burnishing using an indenter with a radius of *R* = 2 mm at a sliding speed of *v_s_* = 40 m/min; burnishing force of *F_b_* = 175 N; feed rate of *f_b_* = 0.025 mm/rev; and number of passes of *n_p_* = 1, 3, and 5	1	400	–90	–610
	3	460	–90	34
	5	480	–	37

**Table 3 materials-17-05656-t003:** Results of a comparative analysis of AISI 304 phase composition in the original state and after nanostructuring burnishing obtained with Bruker D8 Advanced using Rietveld method.

AISI 304 State	Austenite (%)	Martensite (%)
Original state	96.4	0
*n_p_* = 1	72.8	27.2
*n_p_* = 3	37.0	63.0
*n_p_* = 5	19.2	80.8

**Table 4 materials-17-05656-t004:** Comparative overview of various methods of AISI 304 stainless steel surface hardening.

Method Used	Microhardness, HV/ Nanohardness, GPa	Hardening Depth, µm	COF in Dry Condition	Wear Coefficient, m^3^/(N·m)	References
Nanostructuring burnishing	480 HV_0.05_	85	0.22–0.25	5.83 × 10^−15^	This study
Magnetron co-sputtering (CrSiN)	38 GPa	4.5	0.4	5 × 10^−16^	[2]
Reactive magnetron sputtering (WSiN)	45.4 GPa	2	0.3	2 × 10^−15^	[3]
High-velocity air fuel (TiC-Fe)	887 HV_0.03_	200	0.41	5 × 10^−14^	[6]
Low-temperature plasma carburizing	475 HV_0.05_	3.9–11	0.6	1.2 × 10^−15^	[7]
Nitrogen–carbon glow discharge	3.5 GPa	10–50	0.5–0.7	-	[10]
Sandblasting	250–400 HV_0.05_	50–70	0.72	1.13 × 10^−14^	[11]
Polishing	2.4–3.1 GPa	50	0.4	3.38 × 10^−14^	[12]

## Data Availability

The data presented in this study are available on request from the corresponding author.

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
