# Peer review of "Surface Nanocrystallization and Improvement of the Mechanical and Tribological Properties of AISI 304 Steel Using Multi-Pass Nanostructuring Burnishing"

_materials, 2024, doi:10.3390/ma17225656_

Round 1

Reviewer 1 Report

Comments and Suggestions for Authors

1.      As shown in Figure 5, the nanocrystallite size first increases and then decreases with increasing the sliding speed for 1 pass. Why is that?

2.      The X-ray diffraction pattern of the AISI 304 steel samples before and after treatments should be provided in the manuscript.

3.      Can the authors give an detailed explanation of how to calculate the content of residual austenite g and the integral breadth as shown in Table 1.

4.      Can the multi-pass nanostructuring burnishing method be applied to other types of steels?

5.      How many samples of AISI 304 steel at different sliding speeds are used for calculating the nanocrystallite size distributions as shown in Figure 4?

6.      Any other benefits of multi-pass nanostructuring burnishing method compared to previous reported studies like low-temperature plasma carburizing.

7.      As shown in Figure 7, the authors should explain the meaning of T in the Figure for better understanding.

Reviewer 2 Report

Comments and Suggestions for Authors

The authors studied the hardness of stainless steels with various crystal domain sizes. By controlling the sliding speed, the crystal domain sizes of stainless steels were varied. Appropriate control of the sliding speed made the fine grain sizes. This grain size was related to the hardness of stainless steel. This information is helpful for the readers working on the metal. However, there are some concerns about the introduction and the discussion sections. If the authors faithfully revise the manuscript in accordance with my comments, this study will meet the criteria for the publication in Materials.

Comment 1: What brings the oscillation of microhardness in Figure 2?

Comment 2: Please show the physical mechanism of varying the crystal size? Clarifying this mechanism is important to control the crystal domain size.

Comment 3: Stainless steel is one of the iron compounds. In recent years, iron compounds are attracting much attention as the functional materials: e.g. iron oxide memory, iron silicide detector, etc. The authors cannot ignore the studies about other iron compounds except for stainless steel. As aforementioned, there are important studies: iron oxide memory (Sci. Technol. Adv. Mater. 21, 195 (2020).), iron silicide photovoltaics (AIP Advances 12, 105317 (2022).), etc. In the beginning of the introduction section, the authors should comment on the history of iron compounds including the recommended studies and cite the related references. After the history, I recommend the authors’ paying attention to stainless steel as one of the iron compounds.

Reviewer 3 Report

Comments and Suggestions for Authors

Dear authors,

Thank you for this paper in which is proposed to enhance the mechanical and tribological properties of nanocrystallized surfaces of an austenitic chromium-nickel steel formed by nanostructuring burnishing. The influence exerted by sliding speed and number of indenter passes on the formation of a nanocrystalline structure and on modification of microhardness and residual stresses, texture and tribological properties of an austenitic steel is experimentally established and justified.

I can add a few suggestions:

In Introduction – A centralization of the methods used and proposed in the literature for improving the mechanical characteristics of austenitic steels is useful to be presented in a table.

In Chapter 2 the experimental methods should be better described, to include all procedures. Example, to include experiments and Fig. 1, Fig. 11, tribological tests etc.

In Chapter 3 - Experimental Results and Discussions - it is useful to introduce subchapters for the different experiments and results.

Text:

In Fig. 2-Fig. 4 – notations for np and T could be included, as in Fig. 4.

r.181: a short descriptions of Figure 1a is useful.

r. 216: Figure 1 should be improved.

r.220: The Figure 2 shows …

r.243: In Figure 2, the speed value of 40, 120 and 200 m/min is useful to be noted on the graphs, instead of 2, 3, 4. ; np and T should be included.

r.251: Title of Fig. 3 should be improved.

r.254: Figure 4 illustrates …

r.297: … for np = 3…

r.423: Dates from Table 2 suggest ...

r.486: Title of Figure 12 could be improved (on what distance?)

r.532: To indicate the reference for relationship (4).

r.544: In Figure 16, the specific proposed colours for np could be applied.

Round 2

Reviewer 2 Report

Comments and Suggestions for Authors

Everything was cleared. This study is worth publishing.